

# Biological factors associated with long COVID and comparative analysis of SARS-CoV-2 spike protein variants: a retrospective study in Thailand

Supanchita Kiatratdasakul[1,2], Pirom Noisumdaeng[3,4] and Nattamon Niyomdecha[5]

[1] Graduate Program in Medical Technology, Faculty of Allied Health Sciences, Thammasat University, Rangsit Campus, Pathum Thani Province, Thailand
[2] Department of Immunology, Maharat Nakhon Ratchasima Hospital, Mueang, Nakhon Ratchasima, Thailand
[3] Faculty of Public Health, Thammasat University, Rangsit Campus, Pathum Thani Province, Thailand
[4] Thammasat University Research Unit in Modern Microbiology and Public Health Genomics, Thammasat University, Rangsit Campus, Pathum Thani Province, Thailand
[5] Department of Medical Technology, Faculty of Allied Health Sciences, Thammasat University, Rangsit Campus, Pathum Thani Province, Thailand

Corresponding author
Nattamon Niyomdecha,
nattamon@tu.ac.th

## ABSTRACT

**Background.** Post-acute COVID-19 syndrome (long COVID) refers to the persistence of COVID-19 symptoms or exceptional symptoms following recovery. Even without conferring fatality, it represents a significant global public health burden. Despite many reports on long COVID, the prevalence and data on associated biological factors remain unclear and limited. This research aimed to determine the prevalence of long COVID during the two distinct epidemic periods in Thailand, due to the Delta and Omicron variants of SARS-CoV-2, and to investigate the biological factors associated with long COVID. In addition, the spike protein amino acid sequences of the Delta and Omicron variants were compared to determine the frequency of mutations and their potential biological implications.

**Methods.** A retrospective cross-sectional study was established to recruit confirmed COVID-19 participants at Maharat Nakhon Ratchasima Hospital who had recovered for at least three months and were infected between June 2021 and August 2022. The demographic data and long COVID experience were collected via telephone interview. The biological factors were analyzed through binary logistic regression. The datasets of the SARS-CoV-2 spike protein amino acid sequence of the Delta and Omicron variants in Thailand were retrieved from GIDSAID to determine mutation frequencies and to identify possible roles of the mutations based on published data.

**Results.** Data was collected from a total of 247 participants comprising 106 and 141 participants of the Delta and Omicron epidemic periods, respectively. Apart from the COVID-19 severity and health status, the baseline participant data of the two time periods were remarkably similar. The prevalence of long COVID observed in the Omicron period was higher than in the Delta period (74.5% *vs.* 66.0%). The biological factors associated with long COVID were epidemic variant, age, treatment with symptomatic medicines, and vaccination status. When the spike protein sequence data of the two variants were compared, it was observed that the Omicron variant exhibited a greater quantity of amino acid changes in its receptor-binding domain (RBD) and receptor-binding motif (RBM). The critical changes of the Omicron variant

within these regions had a significant function in enhancing virus transmissibility and host immune response resistance.

**Conclusion**. This study revealed informative data associated with long COVID in Thailand. More attention should be given to long COVID caused by unique virus variants and other biological factors to prepare a healthcare management strategy for COVID-19 patients after recovery.

## INTRODUCTION

Since March 11, 2020, COVID-19 caused by SARS-CoV-2 has been classified as a newly emerging disease by the World Health Organization (*WHO, 2020*), a designation it has held for the last four years. Variants of SARS-CoV-2 have emerged continuously on a global scale. Certain variants have demonstrated effective adaptation in order to infect and spread among humans more easily. Since the beginning of COVID-19 era until the present, Thailand has been encountered with three unique epidemic SARS-CoV-2 variants: alpha (April to early June 2021), Delta (June to November 2021), and Omicron (December 2021 until now). The Department of Disease Control of the Ministry of Public Health in Thailand has documented a cumulative count of 4,758,116 confirmed cases and 34,547 fatalities attributed to COVID-19 as of January 28, 2024 resulting in an accumulated case fatality rate of 0.73% (*DDC, 2024*). Although recovery and a low mortality rate of 3–5% have been observed in the majority of individuals infected with COVID-19, a considerable proportion of patients have experienced long-term complications referred to as long COVID (*Batiha et al., 2022*). The prevalence of long COVID in Thailand is limited. However, few studies have suggested a prevalence range of 30–65% for long COVID (*Wangchalabovorn, Weerametachai & Leesri, 2022*; *Wongsermsin, Chinoraso & Yeekian, 2022*; *Tancharoensukjit, 2023*).

Long COVID or post-acute COVID conditions (PCC) are defined by the presence of diverse clinical symptoms that persist for several months shortly after an acute infection with the emerging respiratory virus named severe acute respiratory syndrome coronavirus 2 (SARS-CoV-2). This long-term impact cannot be diagnosed using a specific biomarker or guideline. The disparity in time-frame for genetic variants of concern (VoC) of SARS-CoV-2 investigations contributed to significant differences in the prevalence data of long COVID. (*Montagne et al., 2020*; *Dennis et al., 2021*; *Moreno-Pérez et al., 2021*; *Sudre et al., 2021*). The ongoing COVID-19 pandemic results in the virus adapting through mutations in its genome, particularly in the gene of the spike glycoprotein. Since this protein plays a critical role in facilitating virus entry and is the primary target of neutralizing antibodies, it was hypothesized that this viral structure is responsible for the occurrence of long COVID (*Theoharides, 2022*). Amino acid changes in the spike glycoprotein of the virus allow it to escape the host immune system and to enhance its transmission and pathogenesis. Consequently, infection with distinct genetic variants of SARS-CoV-2 at different times

may have different effects on the occurrence of long COVID symptoms. Our understanding is that there is no published data that indicated a correlation between viral genetic markers and long COVID outcomes. This is because the majority of researchers believe that long COVID is associated with specific host genetic variants (*Batiha et al., 2022*).

Besides the genetic variants of SARS-CoV-2, long COVID might be connected to several biological host variables. The development of long COVID has been associated with several characteristics, including the severity of the illness, aging, gender, and rising levels of certain inflammatory markers (*Batiha et al., 2022*). Nevertheless, the data pertaining to the Asian population, particularly Thai, is extremely limited. This study aimed to identify biological risk factors associated with the development of long COVID in Thai, specifically focusing on the SARS-CoV-2 variants and host factors. Furthermore, the amino acid variations found in the spike protein of the SARS-CoV-2 Delta and Omicron variants were compared for their role in pathogenesis and transmission.

## MATERIALS AND METHODS

### Study design and participants

This cross-sectional study collected retrospective data from confirmed COVID-19 cases who acquired the SARS-CoV-2 infection during the Delta and Omicron epidemic periods, as reported by the Department of Medical Science of the Ministry of Public Health in Thailand (*DMSc, 2023*). Data were analyzed to determine the difference in prevalence and characteristics of long COVID symptoms between the two periods and to identify which biological factors were associated with long COVID. A minimum sample size of 206 positive COVD-19 cases was required, which was calculated using the n4Studies application (https://www.facebook.com/n4Studies/) based on an estimation infinite population proportion method by setting a prevalence difference proportion of 16% (*ONS, 2022*), and an alpha error of 0.05. Based on the medical records of the confirmed SARS-CoV-2 RT-qPCR positive cases at Maharat Nakhon Ratchasima Hospital from June to November 2021 (Delta period) and from December 2021 to the first week of August 2022 (Omicron period), all confirmed COVID-19 cases who were ≥ 18 years old, infected by SARS-CoV-2 for the first time within these periods, and had recovered from the infection for a minimum of three months were included as eligible participants. They were approached at random *via* telephone between March and April 2023 and requested to provide verbal consent for a single remote interview *via* telephone. To reduce the recall bias that influenced the validity assessment, we conducted a primary evaluation of the memorization participants by asking the time period of their infections. Fortunately, all participants were able to recall the disease period. Individuals who were pregnant during the COVID-19 infection, missed contact, had no recollection of long COVID experiences, passed away prior to interview, or lacked the ability to communicate in Thai were excluded from the study.

A peer-evaluated questionnaire was employed to record and analyze retrospective self-report data from each participant during the interview process. The self-reported questionnaire asked about basic demographic data and the experience of COVID-19

disease, including COVID-19 severity, the presence of long COVID symptoms, drug(s) for treatment, and vaccine status. According to the World Health Organization (*WHO, 2022a*), the definition of long COVID in this study is the persistence of existing infection symptoms or the emergence of new symptoms within three months of the initial SARS-CoV-2 infection. These symptoms must persist for a minimum of two months without any other explanation. In this investigation, the spectrum of long COVID disorders is categorized into seven systems, including cardiovascular: chest pain, fast-beating or heart palpitations; neurology: headaches, lightheadedness, pins-and-needles feelings, change in smell or taste, difficulty thinking or concentrating; dermatology: hair loss, rash; psychology: depression or anxiety, sleep problems; respiratory: difficulty breathing or shortness of breath, coughing; generalized symptoms of fatigue and myalgia; and others: diarrhea, stomach pain, constipation, *etc.*

The questionnaire for COVID-19 severity was evaluated by employing the specific criteria classification explanation, which classified the condition as mild, moderate, or severe according to the NIH categories (*NIH, 2022*). Participants were asked regarding the drug(s) they used for treatment in the five most used categories in Thailand, which include Andrographolide herbal medicine, Flavipiravir, Molnupiravir, symptomatic medicines (any medical therapy of a disease that only affects its symptoms, not the underlying cause, such as antipyretic medicine, cold medicine, cough medicine, *etc.*), and steroids and others.

This study was approved by the Ethical Committee of the Maharat Nakhon Ratchasima Hospital Institutional Review Board (MNRH IRB), Ministry of Public Health, Thailand (Certificate No. 139/2022). Participant identifiers were removed from the data and replaced with a participant code. The collected data will be destroyed immediately after publication.

## Comparing genetic variations in the spike gene between the Delta and Omicron variants

The Global Initiative for Sharing Avian Influenza Data (GISAID) database (https://gisaid.org) is a free, open-access database that is available online. Its purpose is to facilitate the rapid exchange of data from key pathogens, such as influenza, SARS-CoV-2, respiratory syncytial virus (RSV), Mpox virus, and arboviruses such as chikungunya, dengue, and Zika. The database also offers genetic sequence and related clinical and epidemiological data associated with human viruses, as well as geographical and species-specific data associated with avian and other animal viruses, to further the informative data on the evolution and transmission of viruses during epidemics and pandemics.

In this study, GISAID database was accessed for datasets of Delta and Omicron SARS-CoV-2 variants collected in Thailand between July 1, 2021 and August 31, 2022. We collected only complete sequence datasets, which included 206 Delta variant sequences and 292 Omicron sequences. These genetic sequences are derived from the study objectives, who have different backgrounds in terms of gender, clinical status, and living areas in Thailand. Mutations causing amino acid changes within the spike protein of the two variants were examined to compare the percentage of accumulated mutation frequency. Subsequently, key mutations within each variant defined as having a difference in the percentage of accumulated mutation frequency over 50% were reviewed to determine

whether their roles associated with pathogenesis, transmission, and immune evasion of the SARS-CoV-2 variant might help to explain a long COVID outcome.

## Statistical analysis

All data were analyzed using the Statistical Package for the Social Sciences (Version 25.0; SPSS Inc., Chicago, IL, USA). Calculation of frequency, percentage, mean and standard deviation was used to characterize the baseline demographic data, to compare the data of long COVID between the Delta and Omicron epidemic periods, and to compare the genetic variations in the spike protein of the SARS-CoV-2 variants. A binary logistic regression test with enter method was used to determine the association between biological factors and long COVID outcome. The strength of association was determined using an odds ratio at 95% confidence interval and a $P$-value $< 0.05$ was considered as significant association. Furthermore, the model summary or goodness of fit derived from a binary logistic regression analysis was elucidated by the values of -2Log Likelihood (-2ll) and Nagelkerke $R^2$ (Pseudo $R^2$).

## RESULTS

### Demographic characteristics

Between June 2021 and August 2022, during the Delta and Omicron epidemic periods in Thailand, a total of 490 confirmed COVID-19 cases at Maharat Nakhon Ratchasima Hospital were randomly asked for informed consent to participate in a telephone interview. A total of 251 subjects enrolled voluntarily; however, 247 were selected for analysis based on the defined inclusion and exclusion criteria. The number of participants who were infected by SARS-CoV-2 during the Delta and Omicron epidemic periods was 106 and 141, respectively. The flowchart of the subject enrollment process is shown in Fig. 1.

The baseline demographics and clinical characteristics of confirmed COVID-19 cases of the Delta and Omicron epidemic periods were compared are shown in Table 1. The mean age of the participants was $40.75 \pm 14.20$ years during Delta and $35.61 \pm 11.70$ years during Omicron. Female gender, age range of 18 to 29 years, obesity, and blood group O were the most prevalent characteristics of the participants in both epidemic periods. Nevertheless, during the Delta period, more than half of the cases ($n = 54$, or 50.9%) involved immunocompromised hosts who had at least one underlying disease. Hypertension, diabetes, hypercholesterolemia, asthma, and allergies were common underlying conditions. In contrast, infection with SARS-CoV-2 during the Omicron period primarily affected healthy hosts ($n = 125$, 88.7%). Almost one-third of the Delta cases ($n = 38$, 35.8%) exhibited symptoms classified as moderate to severe, whereas the Omicron cases predominantly presented mild symptoms. The antiviral drug Favipiravir and symptomatic drug usage was the key treatment utilized by the participants during both variant periods. The majority of the participants affected during the Delta period had not yet been immunized against COVID-19 ($n = 63$, 59.4%). Conversely, a significant numbers of participants affected during the Omicron period had received at least three vaccinations ($n = 80$, 56.7%). In respect to the vaccine type, the inactivated vaccine was predominant during the Delta period ($n = 24$, 55.8%), whereas the mRNA vaccine was
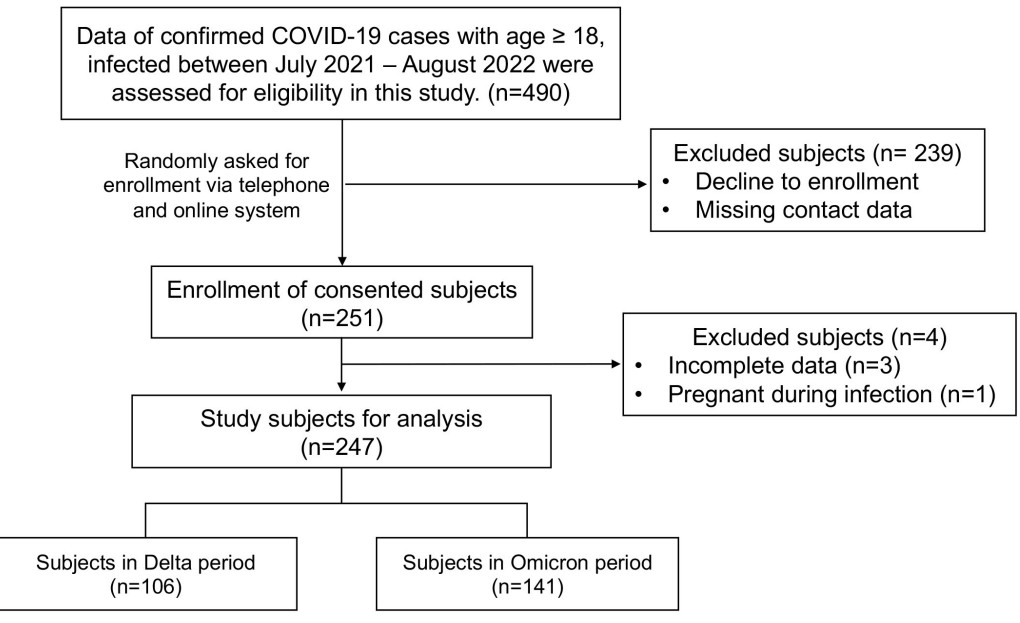

**Figure 1  Flowchart of subject enrollment process and study subject groups.**

predominant during the Omicron period ($n = 75$, 56.4%). Significantly, throughout both epidemic periods, the majority of participants reported having sustained long COVID, with a number of symptoms having comparable prevalence rates between the Delta and Omicron periods.

The prevalence of self-reported long COVID symptoms between the Delta and Omicron epidemic periods was compared (Fig. 2). The seven most frequently reported symptoms in long-term COVID patients were as follows: cardiovascular, neurology, dermatology, psychology, respiratory, generalized symptoms of fatigue and myalgia, and others. Both participant groups reported all symptoms but the rate of observation for each symptom varied between the groups. Participants who were infected during the Delta period showed a predominant psychological disorder, including insomnia, anxiety, and depression. Conversely, abnormalities in the respiratory and neuron systems were prevalent during the Omicron period.

A comparison of the baseline demographics of the participants between non-long COVID cases and long COVID cases is presented in Table 2. Non-long COVID cases had a mean age of 40.86 ± 13.74 years and long COVID cases were 36.56 ± 12.59 years old. Comparative frequencies were observed for almost all analyzed variables in the two groups. Significantly, the prevalence of long COVID experience was greater during the Omicron period (60%) compared to the Delta period (40%); additionally, a greater proportion of participants with moderate to severe COVID-19 exhibited a higher rate of long COVID.

A binary logistic regression analysis was used to determine biological factors associated with long COVID. Variant epidemic period, age, treatment with symptomatic medicines, and vaccination status were found to be significant predictors of long COVID outcomes.

**Table 1  Baseline demographics and clinical details of confirmed COVID-19 cases compared between the Delta and Omicron epidemic periods in Thailand ($n = 247$).**

| Variable | Delta ($n = 106$) | Omicron ($n = 141$) |
|---|---|---|
| **Age, years** | $40.75 \pm 14.20$ | $35.61 \pm 11.70$ |
| 18–29 | 29 (27.4%) | 49 (34.8%) |
| 30–39 | 28 (26.4%) | 47 (33.3%) |
| 40–49 | 23 (21.7%) | 24 (17.0%) |
| more than 50 | 26 (24.5%) | 21 (14.9%) |
| **Sex** | | |
| Male | 36 (34.0%) | 67 (47.5%) |
| Female | 70 (66.0%) | 74 (52.5%) |
| **BMI (kg/m$^2$)** | $25.21 \pm 4.60$ | $24.08 \pm 5.09$ |
| **BMI weight status**[a] | | |
| Underweight (<18.5) | 3 (2.8%) | 15 (10.6%) |
| Normal (18.5–22.9) | 30 (28.3%) | 47 (33.3%) |
| Overweight (23–24.9) | 27 (25.5%) | 29 (20.6%) |
| Obese ($\geq$25) | 46 (43.4%) | 50 (35.5%) |
| **Blood group** | | |
| A | 12 (11.3%) | 20 (14.2%) |
| B | 27 (25.5%) | 29 (20.6%) |
| AB | 10 (9.4%) | 13 (9.2%) |
| O | 35 (33.0%) | 47 (33.3%) |
| Unknown | 22 (20.8%) | 32 (22.7%) |
| **Having underlying diseases** | | |
| Yes | 54 (50.9%) | 16 (11.3%) |
| No | 52 (49.1%) | 125 (88.7%) |
| **No. of underlying diseases** | | |
| 1 | 35 (64.8%) | 16 (100.0%) |
| $\geq$2 | 19 (35.2%) | 0 (0.0%) |
| **Key underlying diseases** | | |
| Hypertension | 29 (27.4%) | 1 (0.7%) |
| Diabetes | 29 (27.4%) | 1 (0.7%) |
| Heart disease | 2 (1.9%) | 1 (0.7%) |
| Cerebrovascular disease | 1 (0.9%) | 0 (0.0%) |
| Thyroid disease | 2 (1.9%) | 3 (2.1%) |
| Asthma and Allery | 9 (8.5%) | 4 (2.8%) |
| Cancer | 2 (1.9%) | 0 (0.0%) |
| Hypercholesterolemia | 12 (11.3%) | 0 (0.0%) |
| **Disease severity**[b] | | |
| Mild | 68 (64.2%) | 132 (93.6%) |
| Moderate | 28 (26.4%) | 9 (6.4%) |
| Severe-ICU | 10 (9.4%) | 0 (0.0%) |

**Table 1** (*continued*)

| Variable | Delta ($n = 106$) | Omicron ($n = 141$) |
|---|---|---|
| **Key drugs for treatment** | | |
| Andrographolide | 15 (14.2%) | 27 (19.1%) |
| Flavipiravir | 61 (57.5%) | 58 (41.1%) |
| Molnupiravir | 1 (0.9%) | 9 (6.4%) |
| Symptomatic medicines[c] | 46 (43.4%) | 72 (51.1%) |
| Steroid and others | 7 (6.6%) | 2 (1.4%) |
| **Vaccination status** | | |
| Unvaccinated | 63 (59.4%) | 8 (5.7%) |
| 1–2 doses | 39 (36.8%) | 53 (37.6%) |
| ≥3 doses | 4 (3.8%) | 80 (56.7%) |
| **Type of the latest vaccine** | | |
| Inactivated | 24 (55.8%) | 15 (11.3%) |
| Viral vector | 15 (34.9%) | 43 (32.3%) |
| mRNA | 4 (9.3%) | 75 (56.4%) |
| **Long COVID status** | | |
| Yes | 70 (66.0%) | 105 (74.5%) |
| No | 36 (34.0%) | 36 (25.5%) |
| **No. of long COVID symptoms** | | |
| 1 | 29 (27.4%) | 41 (29.1%) |
| 2 | 21 (19.8%) | 28 (19.9%) |
| 3 | 9 (8.5%) | 18 (12.8%) |
| ≥4 | 11 (10.4%) | 18 (12.8%) |

**Notes.**

Data reported as n (%) for categorical variables, and mean ± SD for continuous variables.

[a] Based on *Weir & Jan (2023)*.

[b] Based on *WHO (2022a)*.

[c] Any form of medical drug that only targets the symptoms of a disease, such as antipyretic medicine, cold medicine, cough medicine, *etc.*, rather than the underlying cause.

Abbreviations: BMI, Body Mass Index; No, Number.

The odds of developing long COVID were significantly higher for SARS-CoV-2 infection during the Omicron period [OR = 2.976 (95% CI [1.202–7.365]); $p = 0.018$] than for infection during the Delta period. A 1-year increase in age was correlated with a decreased likelihood of developing long COVID [OR = 0.969 (95% CI [0.942–0.996]); $p = 0.025$]. In comparison to other antiviral and immunosuppressive drugs, treatment with symptomatic medications for COVID-19 was associated with a higher risk of developing long COVID [OR = 3.804 (95% CI [1.149–12.590]); $p = 0.029$]. It is worth mentioning that in comparison to vaccination with at least one dose, non-vaccination was a significant risk factor for long COVID [OR = 6.434 (95% CI [1.253–33.033]); $p = 0.026$]. All ten independent variables, as shown in Table 2, possessed the potential to account for 19.3% of the long COVID outcome explanation. The remaining 80.7% of the variance could be accounted for by variables that were not included in the binary logistic regression analysis. The dataset for analysis in this part was available in File S1.

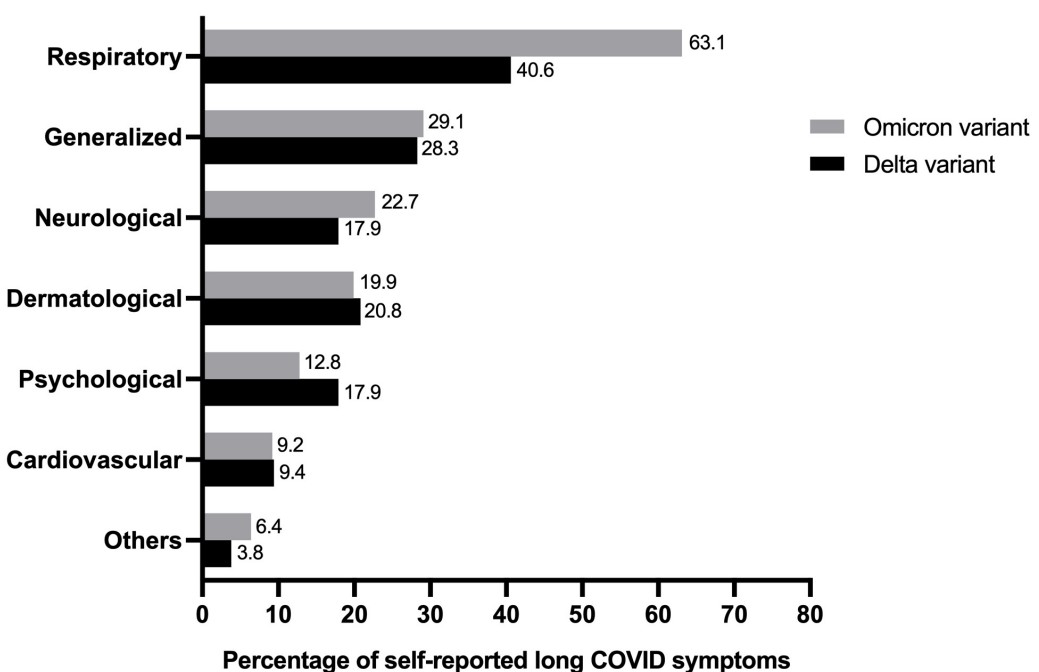

**Figure 2** Comparing percentage of individual's experience of self-reported long COVID symptoms from study population in Delta and Omicron variant periods (total $n = 175$).

## Genetic variations in the spike gene between the Delta and Omicron variants isolated in Thailand

As it has been hypothesized that the spike protein of SARS-CoV-2 is accountable for long COVID, particularly neuropsychiatric symptoms (*Theoharides, 2022*), the prevalence and characteristics of long COVID outcomes may be influenced differently by the mutations in this protein found in the Delta and Omicron variants. This analysis compared the mutation frequencies observed in each region of the spike protein across the two variants as shown in Fig. 3. Most amino acid changes based on the wild-type spike protein sequence were observed in the S1 subunit compared to the S2 subunit. A higher number of amino acid changes were found in nearly all regions of the S1 subunit of the Omicron variant in comparison to the Delta variant, except for the signal peptide (SP) region. The amino acid substitutions observed in the fusion peptide (FP) and heptad repeat 1 (HR1) region of the S2 subunit were similar between the two variants. No amino acid changes were observed in the cytoplasmic domain (CD) and protease cleavage site of the S2 subunit (S2′) for the Omicron variant.

The frequency of each mutation in the S1 and S2 spike protein subunits was compared between the two variants and is presented as heatmap data in Figs. 4 and 5. The predominant high-frequency mutations were found in the S1 subunit of both variants, especially in the regions of the N-terminal domain (NTD), receptor-binding domain (RBD), receptor-binding motif (RBM), and protease cleavage site between the S1 and S2 subunits (S1S2′).

**Table 2** Baseline demographics of study populations comparing between non-long COVID and long COVID cases and variables associated with the risk of long COVID status ($n = 247$).

| Variable | Frequency n (%) or mean ± SD | | Binary logistic regression | | |
|---|---|---|---|---|---|
| | Non-long COVID ($n = 72$) | Long COVID ($n = 175$) | B | Odds ratio (95% CI) | P-value |
| **Variant period** | | | | | |
| Delta | 36 (50.0%) | 70 (40.0%) | – | 1 | – |
| Omicron | 36 (50.0%) | 105 (60.0%) | 1.090 | 2.976 (1.202–7.365) | 0.018[*] |
| **Age, years** | 40.86 ± 13.74 | 36.56 ± 12.59 | −0.032 | 0.969 (0.942–0.996) | 0.025[*] |
| **Sex** | | | | | |
| Female | 42 (58.3%) | 102 (58.3%) | – | 1 | – |
| Male | 30 (41.7%) | 73 (41.7%) | −0.449 | 0.638 (0.304–1.341) | 0.236 |
| **BMI** | 25.03 ± 5.05 | 24.37 ± 4.87 | 0.003 | 1.003 (0.905–1.111) | 0.960 |
| **Blood group** | | | | | |
| O | 22 (41.5%) | 60 (42.9%) | – | 1 | – |
| A | 9 (16.9%) | 23 (16.4%) | −0.209 | 0.812 (0.306–2.153) | 0.675 |
| B | 16 (30.2%) | 40 (28.6%) | −0.132 | 0.876 (0.397–1.936) | 0.744 |
| AB | 6 (11.3%) | 17 (12.1%) | 0.510 | 1.666 (0.553–5.018) | 0.364 |
| **Underlying diseases** | | | | | |
| No | 48 (66.7%) | 129 (73.7%) | – | 1 | – |
| Yes | 24 (33.3%) | 46 (26.3%) | −0.365 | 0.694 (0.213–2.267) | 0.546 |
| Hypertension | 14 (19.4%) | 16 (9.1%) | −1.161 | 0.313 (0.084–1.164) | 0.083 |
| Diabetes | 7 (9.7%) | 9 (5.1%) | −0.803 | 0.448 (0.114–1.757) | 0.249 |
| **Disease severity** | | | | | |
| Mild | 64 (88.9%) | 136 (77.7%) | – | 1 | – |
| Moderate | 7 (9.7%) | 30 (17.1%) | 0.994 | 2.701 (0.948–7.697) | 0.063 |
| Severe-ICU | 1 (1.4%) | 9 (5.1%) | 2.070 | 7.925 (0.768-81.786) | 0.082 |
| **Key drugs for treatment**[a] | | | | | |
| Andrographolide | 11 (15.3%) | 31 (17.7%) | 0.340 | 1.405 (0.510–3.867) | 0.511 |
| Flavipiravir | 35 (48.6%) | 84 (48.0%) | 0.901 | 2.463 (0.670–9.057) | 0.175 |
| Molnupiravir | 5 (6.9%) | 5 (2.9%) | −0.183 | 0.832 (0.128–5.402) | 0.848 |
| Symptomatic medicines[b] | 30 (41.7%) | 88 (50.3%) | 1.336 | 3.804 (1.149–12.590) | 0.029[*] |
| Steroid and others | 1 (1.4%) | 8 (4.6%) | 0.609 | 1.839 (0.186–18.166) | 0.602 |
| **Vaccination status** | | | | | |
| Vaccinated | 51 (70.8%) | 125 (71.4%) | – | 1 | – |
| Unvaccinated | 21 (29.2%) | 50 (28.6%) | 1.862 | 6.434 (1.253–33.033) | 0.026[*] |
| **Type of the latest vaccine** | | | | | |
| Inactivated | 11 (21.6%) | 28 (22.4%) | – | 1 | – |
| Viral vector | 19 (37.3%) | 39 (31.2%) | −0.083 | 0.921 (0.331–2.563) | 0.874 |
| mRNA | 21 (41.2%) | 58 (46.4%) | 0.250 | 1.284 (0.395–4.173) | 0.677 |
| [c]-2Log Likelihood (-2ll) = 262.136 $R^2 = 0.193$ | | | | | |

**Notes.**

[a]Reference group was non-selection in each drug category to treat.

[b]Any form of medical drug that only targets the symptoms of a disease, such as antipyretic medicine, cold medicine, cough medicine, *etc.*, rather than the underlying cause.

[c]To elucidate the model summary or goodness of fit following binary logistic regression analysis, the -2Log Likelihood (-2ll) and Nagelkerke $R^2$ (Pseudo $R^2$) statistic values were reported.

[*]Significant level at *P*-value < 0.05.
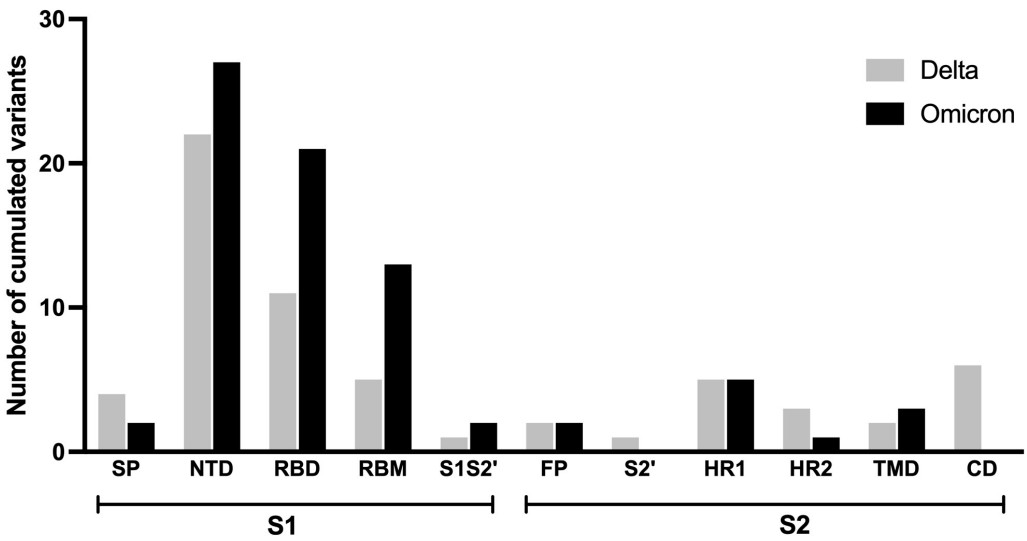

**Figure 3 Comparing number of accumulated mutations between Delta and Omicron variant isolates in S1 and S2 subunits of spike SARS-CoV-2.** SP, signal peptide; NTD, N-terminal domain; RBD, receptor-binding domain; RBM, receptor-binding motif; S1S2′, protease cleavage site between S1 and S2 subunits; FP, fusion peptide; S2′, protease cleavage site in S2 subunit; HR1, heptad repeat 1; HR2, heptad repeat 2; TMD, transmembrane domain; CD, cytoplasmic domain.

For the S2 subunit, high-frequency mutations were observed in the FP region of the Omicron variant and the HR1 region of both variants.

Finally, the literature review of the pathogenesis evidence or suspected evidence regarding the role of the key mutations of each variant, defined here as percentage frequency differences between variants greater than 50%, is shown in Table 3. The Omicron variant sequences obtained from Thailand had a larger number of key mutations compared to the Delta variant sequences. Most of the significant mutations identified in the Delta variant sequences were associated with ACE2 affinity binding enhancement, viral infectivity, and immune evasion. The sole mutation, D950B, located in the HR1 region of the S2 subunit, lacked any evidence regarding its function. The greater part of the key mutations in the Omicron variant exhibited comparable functions to those observed in the Delta variant. However, a few mutations played addition roles in facilitating spike cleavage, virus fusion, and virus transmissibility; spike function impairment had also been published. Additionally, the function of five mutations in the NTD and RBD regions of the S1 subunit remained unknown: L24del, P25del, P26del, A27S, and R408S. The dataset for analysis in this part was available in tFile S2.

## DISCUSSION

The potential of a high public health burden due to long COVID must be considered amid the ongoing global transmission of SARS-CoV-2 and its rapidly emerging variants. Research comparing the variations in long COVID prevalence and characteristics caused by distinct virus variants is scarce. Thus, this study collected informative data on long

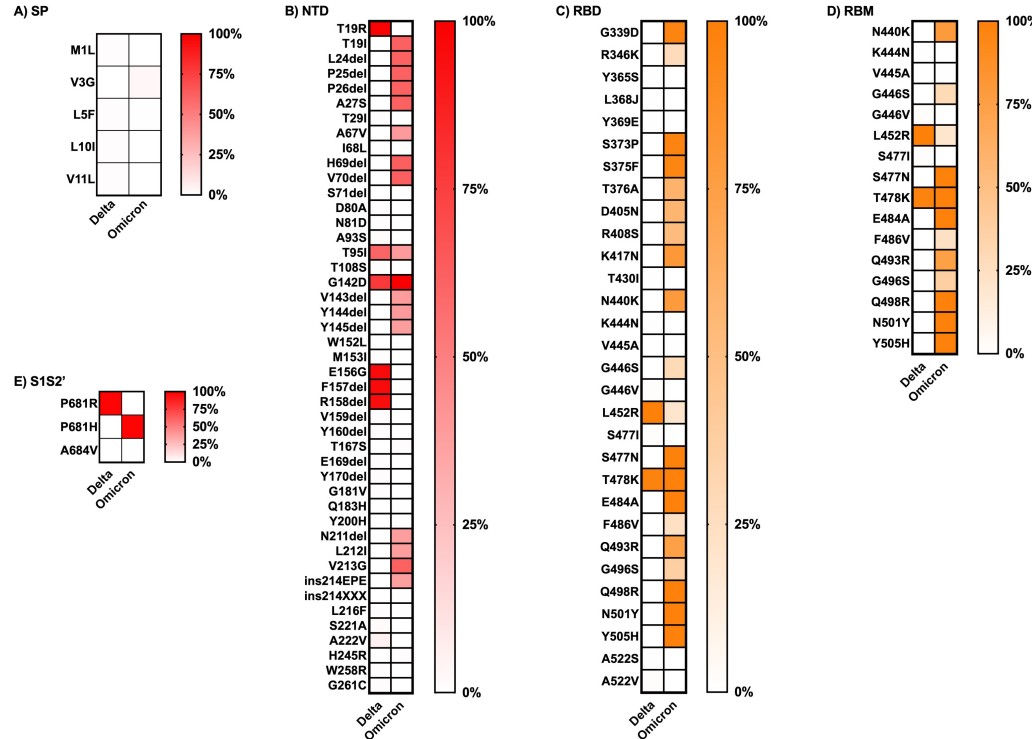

**Figure 4 Comparing the percentage of frequency mutations between Delta and Omicron variant isolates in the S1 subunit of spike SARS-CoV-2.** (A) SP: signal peptide; (B) NTD: N-terminal domain; (C) RBD: receptor-binding domain; (D) RBM: receptor-binding motif; and (E) S1S2′: protease cleavage site between S1 and S2 subunits.

COVID prevalence and clinical symptoms by comparing the Delta and Omicron epidemic periods in Thailand.

Our demographic analysis was consistent with previous studies indicating that most participants in both epidemic periods were female (*Antonelli et al., 2022*; *WHO, 2022b*). The Omicron variant had a higher potential for infection among healthy people compared to the preceding Delta variant. This may be correlated with the biological characteristic of an increased transmissibility of Omicron. It must be emphasized that all strains of SARS-CoV-2 are capable of inducing long COVID. This is despite the fact that the prevalence of long COVID varied considerably between studies due to dissimilarities in the COVID-19 participant selection process, study sites, ethnicity, self-report bias, and time frame analysis (*Antonelli et al., 2022*; *Du et al., 2022*). Over sixty percent of our two participant groups had long COVID. Long COVID prevalence in the Delta period found in this study was comparable with a previous report from Thailand (*Wangchalabovorn, Weerametachai & Leesri, 2022*). However, the data collected from participants' self-reports may be influenced by confounding factors, especially recall bias in the memorization of events that occurred a long time ago. Although this study decreased this bias by employing a question to assess the memory of participants, it may not entirely prevent it.

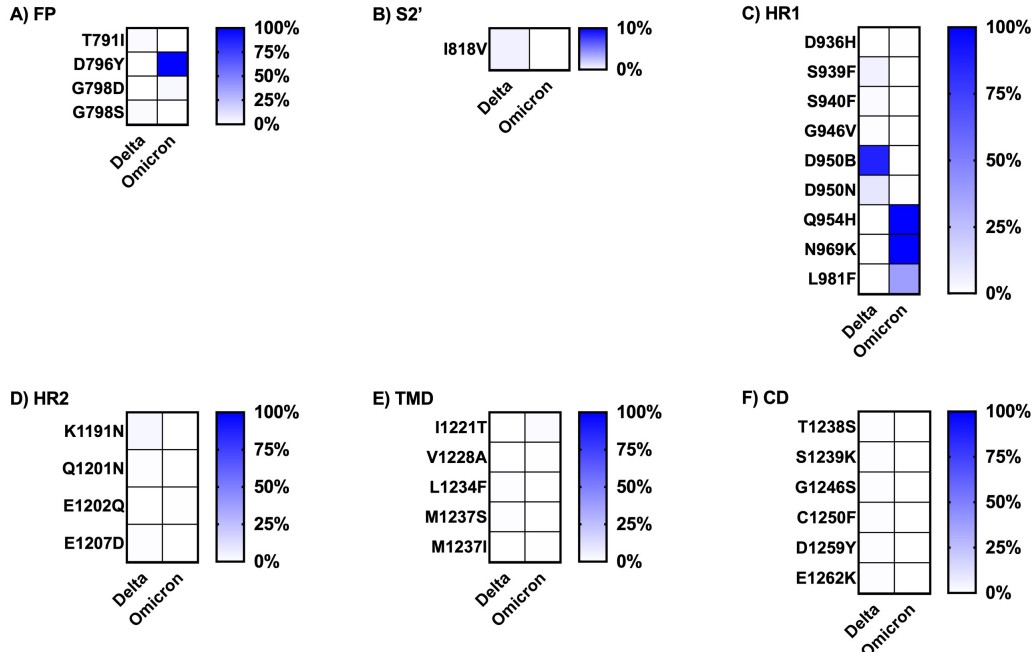

**Figure 5** Comparing the percentage of frequency mutations between Delta and Omicron variant isolates in the S2 subunit of spike SARS-CoV-2. (A) FP: fusion peptide, (B) S2′: protease cleavage site in S2 subunit, (C) HR1: heptad repeat 1, (D) HR2: heptad repeat 2, (E) TMD: transmembrane domain, and (F) CD: cytoplasmic domain.

Post-acute COVID-19 syndrome or casually long COVID comprises a wide spectrum of clinical symptoms as described in this study. Even though the Delta and Omicron groups presented long COVID characteristics at varying rates, all abnormality classifications were discernible in both groups. As reported in studies from Europe and Asia (*Menges et al., 2021*; *Xiong et al., 2021*), fatigue that was categorized as a generalized symptom and respiratory symptoms remained prevalent. Although the precise mechanisms underlying the development of long COVID remain unknown, several hypotheses have been proposed, including immune dysregulation, persistent inflammatory reactions, antibody-dependent enhancement (ADE) from non-neutralizing antibody response, autoimmune mimicry, viral persistence, reactivation of latent pathogens, and alterations in the host microbiome (*Batiha et al., 2022*; *Chen et al., 2023*). With respect to viral persistence and shedding, the upper respiratory tract, lower respiratory tract, gastrointestinal tract, and blood were observed to have the longest durations of viral RNA shedding at 83, 59, 126, and 60 days, respectively (*Chen et al., 2023*). The sustained presence of SARS-CoV-2 RNA or antigen and immune responses directed against it may determine the development of long COVID (*Files et al., 2021*; *Chen et al., 2023*).

In this study, the biological factors that were found significantly associated with the prognosis of long COVID were variant epidemic period, age, treatment with symptomatic medicines, and vaccination status as determined by binary logistic regression analysis. Infection during the Omicron period was associated with an almost three times increased

**Table 3  The frequency difference of mutations in spike SARS-CoV-2 protein between Delta and Omicron variant isolates has a value greater than 50%.**

| Region | Mutation | Accumulated frequency (%) | | % Δ | Pathogenesis or suspected evidence (x) | | | | | | | | Reference |
|---|---|---|---|---|---|---|---|---|---|---|---|---|---|
| | | Delta (N = 206) | Omicron (N = 292) | | Not found | Immune escape | Involve in infectivity | Enhance binding affinity to ACE2 receptor | Enhance viral fusion | Enhance spike clavage | Enhance transmissibility | Impair spike function | |
| **A). S1 subunit** | | | | | | | | | | | | | |
| **NTD** | T19R | 203 (98.5) | 0 (0.0) | 98.5 | | x | | | | | | | *Singh et al. (2022)* |
| | T19I | 0 (0.0) | 180 (61.6) | 61.6 | | | x | | | | | | *Pastorio et al. (2022)* |
| | L24del | 0 (0.0) | 179 (61.3) | 61.3 | x | | | | | | | | |
| | P25del | 0 (0.0) | 179 (61.3) | 61.3 | x | | | | | | | | |
| | P26del | 0 (0.0) | 178 (60.9) | 60.9 | x | | | | | | | | |
| | A27S | 0 (0.0) | 179 (61.3) | 61.3 | x | | | | | | | | |
| | H69del | 0 (0.0) | 181 (61.9) | 61.9 | | | x | | | | | | *Fatihi et al. (2021)* |
| | V70del | 0 (0.0) | 182 (62.3) | 62.3 | | | x | | | | | | *Fatihi et al. (2021)* |
| | E156G | 197 (95.6) | 0 (0.0) | 95.6 | | x | x | x | | | | | *Dang, Ren & Wang (2022)* |
| | F157del | 197 (95.6) | 1 (0.3) | 95.2 | | x | x | x | | | | | *Dang, Ren & Wang (2022)* |
| | R158del | 195 (94.7) | 1 (0.3) | 94.3 | | x | x | x | | | | | *Dang, Ren & Wang (2022)* |
| | V213G | 0 (0.0) | 179 (61.3) | 61.3 | x | | | | | | | | *Wang et al. (2023)* |
| **RBD** | G339D | 0 (0.0) | 282 (96.6) | 96.6 | | x | x | | | | x | | *Cao et al. (2022)* |
| | S373P | 0 (0.0) | 281 (96.2) | 96.2 | | x | x | | | | x | | *Cao et al. (2022)* |
| | S375F | 0 (0.0) | 280 (95.9) | 95.9 | | x | x | | | | x | | *Cao et al. (2022)* |
| | T376A | 0 (0.0) | 173 (59.2) | 59.2 | | | x | | | | | x | *Pastorio et al. (2022)* |
| | D405N | 0 (0.0) | 170 (58.2) | 58.2 | | | x | | | | | x | *Bugatti et al. (2023)* |
| | R408S | 0 (0.0) | 153 (52.4) | 52.4 | x | | | | | | | | |
| | K417N | 0 (0.0) | 235 (80.5) | 80.5 | | x | x | x | | | | | *Greaney et al. (2021)* |
| **RBD/ RBM** | N440K | 0 (0.0) | 230 (78.8) | 78.8 | x | | | | | | | | *Cao et al. (2022)* |
| | L452R | 205 (99.5) | 59 (20.2) | 79.3 | | x | x | x | | | | | *Dang, Ren & Wang (2022), Wilhelm et al. (2021)* |
| | S477N | 0 (0.0) | 287 (98.3) | 98.3 | | | x | x | | | | | *Dang, Ren & Wang (2022), Zahradník et al. (2021)* |
| | E484A | 0 (0.0) | 288 (98.6) | 98.6 | | x | | | | | | | *Greaney et al. (2021)* |
| | Q493R | 0 (0.0) | 218 (74.7) | 74.7 | | x | x | x | | | x | | *Focosi et al. (2021)* |
| | Q498R | 0 (0.0) | 288 (98.6) | 98.6 | | | x | x | | | | | *Dang, Ren & Wang (2022), Zahradník et al. (2021)* |
| | N501Y | 0 (0.0) | 288 (98.6) | 98.6 | | | x | x | | | x | | *Niu et al. (2021)* |
| | Y505H | 0 (0.0) | 288 (98.6) | 98.6 | | | x | x | | | | | *Khan et al. (2022)* |
| **S1S2′** | P681R | 202 (98.1) | 0 (0.0) | 98.1 | | | x | | x | | | | *Saito et al. (2022)* |
| | P681H | 0 (0.0) | 287 (98.3) | 98.3 | | x | x | | | x | | | *Lista et al. (2022)* |
| **B). S2 subunit** | | | | | | | | | | | | | |
| **FP** | D796Y | 0 (0.0) | 289 (98.9) | 98.9 | | x | | | | | | | *Elko et al. (2024)* |
| **HR1** | D950B | 179 (86.9) | 0 (0.0) | 86.9 | x | | | | | | | | |
| | Q954H | 0 (0.0) | 290 (99.3) | 99.3 | | | x | | x | | | | *Park et al. (2023)* |
| | N969K | 0 (0.0) | 290 (99.3) | 99.3 | | | x | | x | | | | *Park et al. (2023)* |

**Notes.**

Symbol and abbreviations: △, different value; x, presence; NTD, N-terminal domain; RBD, receptor-binding domain; RBM, receptor-binding motif; S1S2′, protease cleavage site between S1 and S2 subunits; FP, fusion peptide; HR1, heptad repeat 1.

risk of long COVID sequelae in comparison to the Delta period. This finding contradicts the results of previous studies (*Antonelli et al., 2022*; *Du et al., 2022*). Although the prevalence of long COVID caused by different strains did not differ significantly, *Du et al. (2022)* identified a significant distinction in specific symptoms observed in each strain through a systematic review and meta-analysis. As we utilized a lengthy period during the Omicron epidemic covering the variant epidemics of BA.1, BA.2, BA.4, and BA.5, which contain many critical mutations that increased the fitness for infection (*Tian et al., 2022*), the discrepancy in the period of analysis may account for the outcome of our study. In agreement with the findings of previous studies (*Peghin et al., 2021*; *Maglietta et al., 2022*; *Notarte et al., 2022*; *Subramanian et al., 2022*; *Yoo et al., 2022*), advanced age did not support long COVID induction in this analysis. However, it differed from some studies (*Sudre et al., 2021*; *Thompson et al., 2022*). Based on our current understanding, this study represents the first report that establishes treatment with symptomatic medications and vaccination status as substantial risk factors for the development of long COVID. As long COVID may be caused by the persistence of viral antigen, which results in an ongoing activation of the host immune response, long COVID can be induced by the use of symptomatic medications lacking antiviral activity. Compared to those who had received at least one dose of vaccination, non-immunized individuals exhibited a six-fold increased risk of developing long COVID. This crucial information was substantiated by a recent cohort research (*Catala et al., 2024*). Moreover, our analysis data, along with those of other prior studies, emphasized the absence of a correlation between long COVID development and female gender or initial severity of COVID-19 (*Townsend et al., 2020*; *Simani et al., 2021*; *Al-Kuraishy et al., 2022*; *Al-Thomali et al., 2022*).

The accumulation of mutations in the spike protein gene of SARS-CoV-2 is a significant factor in the emergence of novel variants of concern (VOCs). The Omicron variant exhibited the highest number of mutations compared to earlier VOCs (Alpha, Beta, Gamma, and Delta), with more than 60% of the mutations accumulated in the spike protein gene as opposed to other regions of the genome (*Tian et al., 2022*). Viral infection and transmissibility are determined by amino acid changes in the receptor-binding domain (RBD) and receptor-binding motif (RBM) of the SARS-CoV-2 spike protein. Furthermore, the resistance of the host antibody (Ab) immune response is attributed to the amino acid changes occurring in the spike protein, which serves as the primary target for neutralization (*Tian et al., 2022*). Recently, there has been speculation regarding the potential contribution of the spike protein, either alone or in conjunction with other inflammatory mediators, in the induction of long COVID (*Theoharides, 2022*).

Unfortunately, the amino acid sequence of the spike protein obtained directly from participants with long COVID and those without long COVID in this study could not be accessible for the purpose of explicitly comparing and discussing the viral mutations associated with the development of long COVID. We instead performed a comprehensive analysis by retrieving the available spike protein sequence datasets from GIDSAID. In light of this constraint, a direct prediction of a long COVID induced by a viral mutation is not achievable. The analysis of the amino acid sequence of the spike protein from Delta and Omicron variants in Thailand resulted in insightful findings regarding the frequency of

mutations and the function of critical mutations that occur in these two variants. Consistent with a previous study (*Harvey et al., 2021*), the Omicron variant dataset contained two or more times the number of mutations located in the RBD and RBM compared to the Delta variant dataset. The observed variation in the critical region of the spike protein might support an association between spike protein function and progression of long COVID. The enhanced binding affinity to the angiotensin-converting enzyme-2 (ACE2) receptor, which is specific for SARS-CoV-2 infection, on mast cells may be linked to the development of mast cell activation syndrome (MCAS), one of the underlying mechanisms of long COVID, as hypothesized to result from the amino acid changes in the RBD and RBM regions. Extraordinary symptoms may be induced in long COVID patients due to dysregulation in the release of inflammatory mediators in MCAS (*Batiha et al., 2022*).

## CONCLUSION

The SARS-CoV-2 variant, age, treatment with symptomatic medications, and non-immunization status were identified as biological factors associated with long COVID progression in our retrospective cross-sectional study. The identification of these factors should help to facilitate the development of a suitable health management strategy. Furthermore, the analysis of the frequency and biological implications of mutations observed in the spike protein gene of the Delta and Omicron variants could support the explanation for long COVID development and provide a scientific reference for monitoring, prevention, and vaccine development.

## ACKNOWLEDGEMENTS

The authors would like to thank all participants who enrolled in this study.

### Funding
The authors received no funding for this work.

### Competing Interests
The authors declare there are no competing interests.

### Author Contributions

- Supanchita Kiatratdasakul performed the experiments, analyzed the data, prepared figures and/or tables, and approved the final draft.
- Pirom Noisumdaeng conceived and designed the experiments, authored or reviewed drafts of the article, and approved the final draft.
- Nattamon Niyomdecha conceived and designed the experiments, performed the experiments, analyzed the data, prepared figures and/or tables, authored or reviewed drafts of the article, and approved the final draft.

## Ethics

The following information was supplied relating to ethical approvals (i.e., approving body and any reference numbers):

This study was approved by the Ethical Committee of the Maharat Nakhon Ratchasima Hospital IRB, Ministry of Public Health, Thailand (Certificate No. 139/2022).

## Data Availability

The raw data are available in the Supplemental Files.

## Supplemental Information

Supplemental information for this article can be found online at http://dx.doi.org/10.7717/peerj.17898#supplemental-information.

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
