# Peer review of "Biological factors associated with long COVID and comparative analysis of SARS-CoV-2 spike protein variants: a retrospective study in Thailand"

_PeerJ, doi:10.7717/peerj.17898_

## Round 0.1 · original submission · Minor Revisions

Dear authors, many thanks for your patience. We have now reunited enough reviews for your manuscript. Please, refer to the reviewer's reports for medium revisions.

Although i considered these minor revisions, I urge you to be thorough and address all the concerns or questions reported. It can take some time and that's ok.

Reviewer 1 ·

Basic reporting

Pass

Experimental design

-

Validity of the findings

Add more following the attached file.

Additional comments

No

Annotated reviews are not available for download in order to protect the identity of reviewers who chose to remain anonymous.

Reviewer 2 ·

Basic reporting

This study offers valuable insights into the prevalence and biological factors linked to long COVID across distinct epidemic periods in Thailand, with a particular focus on the Delta and Omicron variants. The comprehensive approach, incorporating both clinical and genomic analyses, significantly enriches our understanding of long COVID. Employing a retrospective cross-sectional study design and telephone interviews for data collection allows for a thorough examination of long COVID among recovered COVID-19 patients. Additionally, the comparison of spike protein amino acid sequences between the Delta and Omicron variants sheds light on potential biological implications, particularly concerning virus transmissibility and immune response resistance. These findings underscore the importance of considering unique virus variants and other biological factors in healthcare management strategies for COVID-19 patients post-recovery. Overall, this study contributes valuable data to the field and highlights the need for continued research and attention to long COVID in diverse contexts. However, to enhance the quality of the work, I recommend the following revisions:

1. In the Introduction section (line 71-72), provide the prevalence of long COVID among SARS-CoV-2 infected Thailand adults using nationally representative samples to serve as a reference point for readers.

2. In the second paragraph of the Introduction (line 73-86), address literature gaps related to the research question, specifically exploring existing investigations and research gaps regarding mutations of distinct SARS-CoV-2 variants, particularly focusing on data from the Asian population.

Experimental design

1. Clarify the timing of telephone interviews in relation to the confirmed infection/positive test date to assess the severity of recall bias.

2. Define the main outcome of long COVID (line 117) and specify the timing in relation to infection and symptoms considered, referencing established definitions such as those from WHO, CDC, or ONS.

3. Clarify how COVID-19 severity was measured (line 116) and whether surveillance data (where the infection data was extracted) were used to define COVID-19 severity.

4. Provide details of medication included in the symptomatic medications category (line 117 and table 2).

5. Offer background information on the GISAID database and the characteristics of the study objectives who contributed those 498 samples (line 124-133).

6. Describe the symptoms included in clusters of cardiovascular, neurology, dermatology, psychology, respiratory, and generalized symptoms in the Methods section.

Validity of the findings

1. Line 194-197, you reported that treatment with symptomatic medications for COVID-19 was associated with higher risk of developing long COVID, compared to other drugs. However, in Table 2 footnote item a, the reference group for each drug was non-selection. Please address the discrepancy.

2. In line 186-187, the participants with moderate to severe COVID-19 exhibited a higher rate of long COVID. It is also reported that treatment with symptomatic medications for COVID-19 was associated with higher risk of developing long COVID. It's possible that participants with moderate to severe COVID-19 were more likely to receive symptomatic medications. I recommend examining the correlations between the severity of COVID-19 and the use of symptomatic medications.

3. Please provide the statistical approach you used to assess the outcome in line 199-201 (the 10 dependent variables account for 19.3% of the long COVID outcome explanation).

4. Please include a paragraph of limitations in the Discussion section.

·

Basic reporting

The figures are self explanatory.
It is a nicely drafted manuscript.

Experimental design

The methodology was clear and enough information was provided to replicate the study.
The research question was well defined and relevant.

Validity of the findings

Conclusion is well stated.
Results are statistically sound.

Additional comments

What are the limitations of the study? Please mention it.

Reviewer 4 ·

Basic reporting

The article passes this section.

Experimental design

The article passes this section.

Validity of the findings

The article passes this section.

---

## Round 0.2 · accepted · Accept

Dear authors, thank you for the submission of your interesting work to PeerJ. I am now accepting your manuscript for publication. Congratulations.

Reviewer 2 ·

Basic reporting

The revisions have significantly improved the clarity and quality of the manuscript. The additional data and analyses provided have strengthened the findings and resolved the issues raised in the initial review. The methodology is sound, the results are robust, and the discussion is comprehensive and well-supported by the evidence presented.

Experimental design

NA

Validity of the findings

NA

·

Basic reporting

No comments.

Experimental design

No comments.

Validity of the findings

No comments.

Additional comments

The authors have revised the manuscript based on the suggestions.
Great Job!